



# Tidal resonance in the Gulf of Thailand

Cui Xinmei[1,2], Fang Guohong[1,2], Wu Di[1,2]

[1]First Institute of Oceanography, State Oceanic Administration, Qingdao, 266061, China

[2]Laboratory for Regional Oceanography and Numerical Modelling, Qingdao National Laboratory for Marine Science and
Technology, Qingdao, 266237, China

*Correspondence:* Guohong Fang (fanggh@fio.org.cn)

**Abstract.** The Gulf of Thailand is dominated by diurnal tides, which indicates that the resonant period of the gulf is potentially close to one day. However, when applied to the gulf, the classic quarter wavelength resonant theory fails to give a diurnal resonant period. In this study, we first perform a series of numerical experiments showing that the resonant period of 10 the gulf is approximately one day and that the resonance of the South China Sea body has a critical impact on the resonance of the gulf. In contrast, the resonance of the Gulf of Thailand has little influence on the resonance of the South China Sea body. An idealised two-channel model that can reasonably explain the dynamics of the tidal resonance in the Gulf of Thailand is then established in this study.

## 1 Introduction

The Gulf of Thailand (GOT) is an arm of the South China Sea (SCS), the largest marginal sea of the western Pacific Ocean (Fig. 1). The width of the GOT is approximately 500 km, and the length of the GOT from the top to the mouth of the gulf, as indicated by the red line B in Fig. 1, is approximately 660 km. The mean depth in this area is 36 m according to the ETOPO1 depth data set (from the US National Geophysical Center). The mean depth of the SCS from the southern opening of the Taiwan Strait to the cross section at 1.5 °N is 1323 m. If the GOT is excluded, the mean depth of the rest of the SCS (herein 20 called the SCS body and abbreviated as SCSB) is 1457 m. Tidal waves propagate into the SCS from the Pacific Ocean through the Luzon Strait (LS) and mainly propagate in the southwest direction towards the Karimata Strait, with two branches that propagate northwestward and enter the Gulf of Tonkin and the GOT. The energy fluxes through the Mindoro and Balabac Straits are negligible (Fang et al., 1999; Zu et al., 2008; Teng et al., 2013). The GOT is dominated by diurnal tides, and the strongest tidal constituent is $K_1$ (Aungsakul et al., 2011; Wu et al., 2015).

The resonant responses of the GOT to tidal and storm forcing have attracted extensive research interest. However, the previous results have been diverse. Yanagi and Takao (1998) simplified the GOT and Sunda Shelf as an L-shaped basin and concluded that this basin has a resonant frequency near the semi-diurnal tidal frequency. Sirisup and Kitamoto (2012) applied a normal mode decomposition solver to the GOT and Sunda Shelf area and obtained four eigenmodes with modal periods of 56.61, 20.05, 15.60 and 13.64 h. Tomkratoke et al. (2015) further studied the characteristics of these modes and 30 concluded that the mode with a period of 20.05 h is the most important. Cui et al. (2015) used a numerical method to





estimate the resonant periods of the seas adjacent to China, including the SCS and the GOT, and found that the GOT has a major resonant period of 23.8 h and a minor peak response period of 59.9 h. In these studies, except for Yanagi and Takao (1998), no effort was made to establish a theoretical model of the GOT, and the resonant period estimated by Yanagi and Takao (1998) cannot be used to explain the resonance of the diurnal tides.

The GOT is a semi-enclosed gulf with an amphidromic point in the basin for each $K_1$ and $O_1$ constituent. Wu et al. (2013) reproduced the tidal system well with superimposed incident and reflected Kelvin waves and a series of Poincare modes. This result raises the question of whether the quarter wavelength resonant theory can explain the tidal resonance in the gulf, as is the case with other areas (Miles and Munk, 1961; Garrett, 1972; Sutherland et al., 2005). According to quarter wavelength theory, because the distance from the head of the gulf to the mouth is approximately 660 km and the mean depth

is approximately 36 m, the resonant period should be 39.04 h, which is much longer than the estimates by Cui et al. (2015) and Tomkratoke et al. (2015). Thus, it is clear that the tidal resonance phenomenon in the GOT cannot be reasonably explained by the quarter wavelength theory.

According to the theories of Garrett (1972) and Miles and Munk (1961), tidal oscillations are limited to a specific area. In contrast, Godin (1993) proposed that tides are a global phenomenon that cannot be separated into independent subdomains.

The GOT is an auxiliary area of the SCS and is connected to the SCSB, we thus believe that Godin's theory is applicable to the GOT. In this paper, by considering the bathymetry of the SCSB in numerical experiments and theoretical analyses, we investigate the reasons why the resonant period in the GOT produces an approximately diurnal tide and why the resonant periods of the GOT and SCSB are closely related.

## 2 Numerical methods for estimating the resonant frequency

### 2.1 Governing equations

This study is limited to the main mechanism of diurnal tidal resonance and does not intend to simulate the tides; accordingly, the elimination of tidal forces and nonlinear terms in the control equation will not affect the problem we are studying, and the two-dimensional model effectively suits our purpose (e.g., Garrett, 1972; Godin, 1993; Cui et al., 2015). The general forms of the continuity equation and motion equation used in this study are as follows:

$$\frac{\partial \tilde{\zeta}}{\partial t} = -\frac{1}{R\cos\varphi}\left[\frac{\partial(H\tilde{u})}{\partial\lambda} + \frac{\partial(H\tilde{v}\cos\varphi)}{\partial\varphi}\right], \tag{1}$$

$$\frac{\partial \tilde{u}}{\partial t} = 2\Omega\tilde{v}\sin\phi - \frac{g}{R\cos\varphi}\frac{\partial\tilde{\zeta}}{\partial\lambda} - \frac{\tau\tilde{u}}{H}, \tag{2}$$



$$\frac{\partial \tilde{v}}{\partial t} = -2\Omega \tilde{u} \sin \phi - \frac{g}{R} \frac{\partial \tilde{\zeta}}{\partial \varphi} - \frac{\tau \tilde{v}}{H}, \tag{3}$$

where $t$ denotes time, $\lambda$ and $\phi$, respectively, refer to the east longitude and north latitude; $\tilde{\zeta}$ is the surface height above the undisturbed sea level; $\tilde{u}$ and $\tilde{v}$ represent the east and north components of the fluid velocity, respectively; $R$ indicates the Earth's radius; $\Omega$ refers to the angular speed of the Earth's rotation; $g$ is gravitational acceleration; $H$ denotes the water depth; and $\tau$ represents the linearized bottom friction coefficient.

## 2.2 Open boundary condition

The open boundary condition can be written in the following form:

$$\tilde{\zeta}(i,j,t) = \sum_{n=N_i}^{N_2} \tilde{Z}_n(i,j) \cos[2\pi f_n t - \tilde{\theta}_n(i,j)], \tag{4}$$

where $(i,j)$ are grid points on the open boundary; $f_n = n\Delta f$ refers to the frequency of the $n$-th wave of interest, with $\Delta f$ referring to the spectrum resolution; and for $n = N_1, N_1 + 1, ..., N_2$, $\tilde{Z}_n$ and $\tilde{\theta}_n$ represent the amplitude and phase lag of the $n$-th wave, respectively. In this study we choose $\Delta f = 1/1024$ h$^{-1}$ for the following two reasons: First, the value 1024 is equal to $2^{10}$, enable us to efficiently calculate spectra from model-produced time series by using a fast Fourier transform (FFT). Second, because the minimum frequency difference between the main tidal constituents (Q$_1$, O$_1$, K$_1$, N$_2$, M$_2$, and S$_2$) is equal to $1.51 \times 10^{-3}$ h$^{-1}$, the resolution of $\Delta f = 1/1024$ h$^{-1}$ is sufficient for separating these constituents.

## 2.3 Numerical model of the seas adjacent to China

The computational domain selected is in the range of 99-131 °E and of 1.5-42 °N. The northern and eastern open boundaries are chosen to locate far beyond the area of the SCS to prevent the numerical values at the open boundaries from influencing the results in the study area. The grid resolution is 1/12°. The water depths are basically taken from the ETOPO1 data set and are modified using depth data extracted from navigational charts.

The amplitudes $\tilde{Z}_n(i,j)$ at the open boundaries are specified as a constant, 2 cm; the phase-lags $\tilde{\theta}_n$ are given as random numbers that are evenly distributed in the interval $(0, 2\pi)$ and generated using a normal random number generator. The purpose of using random phase lags is to avoid all or some of the waves to have the same phase at a certain time, which can lead to simultaneous unreasonably high or low sea levels. The selected $N_1$ and $N_2$ values are 1 and 107, respectively.



Thus, the frequencies of the waves studied range from 1/1024-107/1024 h$^{-1}$ or 0.0234-2.5078 day$^{-1}$ (cycles per day). The corresponding periods range from 9-1024 h (approximately 43 days), which covers all main tidal frequencies.

In the last cycle of 1024 hours, the hourly results at each grid point are preserved, and FFT analysis is performed to yield amplitude $Z_n(i,j)$ and phase lag $\theta_n(i,j)$, where $(i,j)$ denotes a grid point in the inner computational domain. The amplitude ratio is defined as follows:

$$G_n(i,j) = Z_n(i,j)/\tilde{Z}_n,$$ (5)

and the phase lag difference is given by the following equation:

$$\psi_n(i,j) = \theta_n(i,j) - \tilde{\theta}_n.$$ (6)

According to Munk and Cartwright (1966), $Ge^{-i\psi}$ is the admittance. Specifically, as Munk and Cartwright (1966) stated, $G_n(i,j)$ is the amplitude response and represents the amplification factor of the $n$-th wave at point $(i,j)$ in

response to forcing. In the present study we call $G_n$ and $\psi_n$ the amplitude gain and phase change, respectively, in accordance with Sutherland et al. (2005) and Roos et al. (2011). In these experiments we use the area-mean value of the top 20% amplitude gains to represent the amplitude gain of the corresponding area.

## 3 Influence of resonance in the South China Sea on the Gulf of Thailand

As mentioned above, the GOT is a semi-enclosed gulf on the continental shelf and can be regarded as an arm of the SCS,

and the rest of the SCS can be regarded as the SCS body (SCSB), which has a mean depth of 1457 m. To examine the influence of SCSB resonance on GOT resonance, we conducted three numerical experiments: Exp. 1 to Exp. 3. In Exp. 1, we used real bottom topography. In Exp. 2, we artificially made the depths in the SCSB equal to half of the real depths and retained the depths in the GOT. In Exp. 3, the depths in the SCSB were artificially doubled, and the depths in the GOT remained unchanged. The results of these three experiments are shown in Table 1 and Fig. 2.

Table 1 and Fig. 2 show that the resonant periods of the response functions of the SCSB and the GOT are 24.36 h and 23.79 h, respectively, when real water depths are used. The resonant periods of the two areas are roughly the same, and both are close to the period of the diurnal tide K$_1$. The peak amplitude gains in these two areas are reduced when the depths of the SCSB are changed to half of the real depths. This reduction in amplitude gain occurs because friction increases as depth decreases (see Eqs. (2) and (3)). Moreover, the resonant periods in the GOT and SCSB both change to 32.00 h. (Fig. 2 and

Table 1). Although both changes are large, there is no difference between the SCSB and GOT resonant periods, indicating that the resonance of the GOT is closely related to that of the SCSB. When the depths of the SCSB are doubled, the resonant periods of the SCSB and GOT are reduced to 14.90 and 15.00, respectively (Fig. 2 and Table 1), again indicating that the





resonance of the GOT is closely related to that of the SCSB. In Exp. 3, there is another weaker peak in the SCSB at the frequency of approximately 1.15 d⁻¹ (Fig. 2a); this result might be caused by the use of two times the real depth in the SCSB,

and the SCSB is deeper than the LS. Moreover, the amplitude gains in the GOT are significantly increased by increasing the depth, which results in reduced friction (see Eqs. (2) and (3)).

In summary, the resonant periods or frequencies change when the depth of the SCSB varies, and the trends in the two sea areas are consistent (Fig. 2). The resonant frequency decreases when the SCSB is shallower and increases when the SCSB is deeper. Additionally, the resonant frequencies of the SCSB and GOT remain almost identical (Table 1). The

experimental results show that the SCSB has a critical impact on the tidal resonance of the GOT and that the GOT is not an independent sea area in terms of tidal resonance.

The above experiments suggest that the tidal resonance in the GOT is strongly affected by the SCSB, the main part of the SCS. Here, we conduct an additional numerical experiment to investigate whether the GOT can also influence the resonance of the SCSB. In this experiment, the boundary of the GOT (indicated by the red line B in Fig. 1) is artificially

closed, and the response function of the SCSB is examined. The results show that the GOT has a small influence on the response of the SCSB, as shown in Fig. 3. When boundary B is closed, the resonant frequency of the SCSB becomes slightly higher than the frequency of the $K_1$ tide, and the response amplitude of the SCSB in the vicinity of the resonant frequency is slightly reduced. As indicated by Arbic et al. (2009), when a shallow basin is connected to a deep basin, the impact of the shallow basin on the tidal response in the deep basin is determined by the depth ratio, width ratio, and length ratio of these

two basins, as well as the friction in the shallow basin. In the present case, the depth and width ratios of the GOT against the SCSB are small, and tides in the GOT are strongly damped, so the impact of the GOT on the SCSB is insignificant.

## 4 A theoretical model

The quarter wavelength resonant theory is based on the wave behaviour in a single channel. As mentioned above, this theory is not applicable to the GOT. Here, we establish a two-channel model and examine its applicability to the SCSB-GOT

system. Tidal waves from the Pacific Ocean propagate through the LS, pass the SCSB and finally enter the GOT. The tidal waves in the Karimata Strait are very weak (Wei et al., 2016) and are not able to propagate into the GOT. Thus, the tidal energy in the GOT is mainly from the SCSB. Therefore, we use a two-channel model to represent the SCSB-GOT system, as shown in Fig. 4. In the figure, $H_1$ is the depth of channel 1 (the deep channel), $H_2$ is the depth of channel 2 (the shallow channel), and $L_1$ and $L_2$ are the lengths of channels 1 and 2, respectively. The tidal waves enter channel 1 through the

opening at $x = L_1$, enter channel 2 through the junction point at $x = 0$, and finally reach the top of channel 2 at $x = -L_2$. The equations governing the tidal motion in channels can be expressed as follows:



$$\begin{cases} \dfrac{\partial \tilde{u}_m}{\partial t} = -g \dfrac{\partial \tilde{\zeta}_m}{\partial x} - \gamma_m \tilde{u}_m \\[2mm] \dfrac{\partial \tilde{\zeta}_m}{\partial t} = -H_m \dfrac{\partial \tilde{u}_m}{\partial x} \end{cases}, \tag{7}$$

where $\tilde{\zeta}_m(x,t)$ and $\tilde{u}_m(x,t)$ represent the elevation and velocity, respectively; $H_m$ is the depth; $\gamma_m$ is the friction

parameter (equivalent to $\tau/H$ in Eqs. (2) and (3)); $g$ is the acceleration due to gravity; and $m = 1, 2$ represents the

different channel segments. Here, $\tilde{u}_m(x,t)$ and $\tilde{\zeta}_m(x,t)$ can be expressed in the forms of $u_m(x,t) = \mathrm{Re}\left(u_m(x)e^{-i\omega t}\right)$

and $\zeta_m(x,t) = \mathrm{Re}(\zeta_m(x)e^{-i\omega t})$, respectively, where $\omega$ is the angular frequency, $t$ is time, and $\zeta_m(x)$ and $u_m(x)$

represent the complex amplitudes of the elevation and velocity, respectively. The boundary and matching conditions are as

follows:

$$\zeta_1(L_1) = a_0, \ \zeta_1(0) = \zeta_2(0), \tag{8}$$

$$H_1 u_1(0) = H_2 u_2(0), \text{ and} \tag{9}$$

$$u_2(-L_2) = 0. \tag{10}$$

From the governing equations and boundary/matching conditions, the complex amplitudes of the elevations of the two

channels can be obtained as follows (see the Appendix for a detailed derivation):

$$\zeta_1(x) = a_0 R_0 Q(x), \text{ and} \tag{11}$$

$$\zeta_2(x) = a_0 R_0 \cos \beta_2 (x + L_2), \tag{12}$$

where

$$R_0 = \cfrac{1}{\cos \beta_1 L_1 \cos \beta_2 L_2 - r \dfrac{p_1}{p_2} \sin \beta_1 L_1 \sin \beta_2 L_2}, \text{ and} \tag{13}$$

$$Q(x) = \cos \beta_1 x \cos \beta_2 L_2 - r \frac{p_1}{p_2} \sin \beta_1 x \sin \beta_2 L_2. \tag{14}$$

In the above equations, $\beta_m = \beta_m' + i\beta_m''$, $\beta_m'^2 = \dfrac{k_m^2(1 + \sqrt{1 + \gamma_m^2})}{2}$, $\beta_m''^2 = \dfrac{k_m^2(-1 + \sqrt{1 + \gamma_m^2})}{2}$, $C_m = \sqrt{gH_m}$,

$\mu_m = \dfrac{\gamma_m}{\omega}$, $k_m = \dfrac{\omega}{C_m}$, and $p_m = (1 + i\mu_m)^{1/2}$. Here $k_m$ is the wavenumber, and $\beta_m$ is equal to $k_m$ in the absence of





friction. Eqs. (11-12) show that the resonant conditions of the two sea areas are consistent. When the denominator of Eq. (13) is equal to zero, the water level of the two-sea area becomes infinitely large, and the condition is as follows:

$$\cos \beta_1 L_1 \cos \beta_2 L_2 = r \left. p_1 \middle/ p_2 \right. \sin \beta_1 L_1 \sin \beta_2 L_2 , \tag{15}$$

where $r = \sqrt{H_2/H_1}$. If $H_2 \ll H_1$ and $\cos \beta_1 L_1 \cos \beta_2 L_2 = 0$, then resonance occurs. $H_2 \ll H_1$ indicates that the

right side of Eq. (15) approaches zero. Based on the configurations of the SCSB and GOT, we take $L_1 = 2600$ km,

$L_2 = 660$ km, $H_1 = 1457$ m, $H_2 = 36$ m, and $\gamma_m = \dfrac{0.0001}{H_m}$.

By substituting these values into (11) and (12), we can obtain the solutions of these equations. The amplitude gains at locations $x = 0$ and $-L_2$ are functions of $\omega$ or functions of the frequency $f$ ( $f = \omega/2\pi$ ) and can be used to represent the response properties of channels 1 and 2, respectively. The results are shown with red curves in Fig. 5. For comparison,

we also calculate the corresponding functions in the absence of friction, as shown by the blue curves in Fig. 5. From these curves, the resonant frequencies can be readily obtained, as given in Table 2.

Figure 5a displays the response function at $x = -L_2$, which represents the response of channel 2. This figure shows that the results in the presence of friction are more realistic than those in the absence of friction. The former case has a maximum peak at a frequency of 1.040 d$^{-1}$. The corresponding resonant period is 23.08 h, and this value is similar to the

result of the numerical experiment involving the natural basin (Table 1, Exp. 1). The secondary response peak appears at a frequency of 0.558 d$^{-1}$, which is also fairly consistent with the results for the natural basin, as shown in Fig. 2b (red curve). The third peak is very small and appears at a frequency of 1.848 d$^{-1}$. If we carefully examine the red curve in Fig. 2b, we can also find a small peak near a frequency of 1.9 d$^{-1}$. The response function is worse when friction is neglected than when friction is retained, but the obtained resonant frequencies are almost unchanged, as shown by the blue curve in Fig. 5a.

Figure 5b shows the response function at $x = 0$, which represents the response of channel 1. The red curve has only one peak at a frequency of 1.040 d$^{-}$, which is also similar to the results of the numerical experiment applied to the natural basin (Table 1, Exp. 1). When the friction is neglected, the frequency of the main peak is unchanged. In addition, there are two other peaks that are very narrow, indicating that these two peaks are relatively insignificant.

Figure 6 displays the amplitude gains along the channels when channel 2 is in a resonant state. In this figure, the

presence/absence of friction is shown in panel a/b. The figure shows that when the forcing frequency is equal to 1.040 d$^{-1}$, which is the major resonant frequency, the amplitude gain gradually increases from the mouth towards the head in channel 1. In channel 2, the amplitude gain decreases first and reaches a trough before increasing again towards the head. The trough corresponds with the amphidromic zone of the diurnal tides in the GOT (see the cotidal charts in Wu et al., 2015). For a frequency of 0.558 d$^{-1}$, the amplitude gain is nearly constant in channel 1 and increases with a relatively high rate towards

the head in channel 2. For a frequency of 1.848 d$^{-1}$, the amplitude gain is also nearly constant in channel 1, and in channel 2,

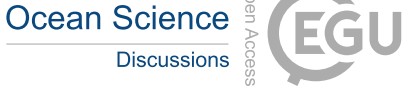

there are two antinodes and one node. This result is similar to the distribution of semi-diurnal tides in the GOT. When friction is neglected, the basic characteristics are the same, but the amplitude gain significantly increases.

**5 Conclusions**

The GOT is dominated by diurnal tides, indicating that the response near the diurnal tide frequency in the GOT is stronger
than that at other frequencies. Our study shows that this feature is mainly related to tidal resonance. However, when applied to the GOT, the classical quarter wavelength resonant theory fails in giving a diurnal resonant period. Changing the water depth in the SCSB in our numerical experiments further shows that the resonance in the GOT is closely related to the resonance in the SCSB. An idealised two-channel model that can reasonably explain the resonance in the GOT is established. However, there are still some problems that require further exploration, such as the effects of the length, width, depth,
Coriolis force and friction of the SCSB on the GOT, which will be the focus of subsequent studies.

**Code availability**

In this paper, we use the Princeton Ocean Model (POM), which is available online at http://www.ccpo.odu.edu/POMWEB/.

**Data availability**

The ETOPO1 data (doi: 10.7289/V5C8276M) are available online at https://www.ngdc.noaa.gov/mgg/global/.

**Appendix: the analytical solution to the tidal wave equation for the two-channel model**

In this appendix we shall give a detailed derivation process for the solution of the two-channel model presented in Section 4. The following symbols shall be used in the derivation:

$i$ : imaginary units

$t$ : time

$\omega$ : angular velocity

$g$ : gravitational acceleration

$\gamma$ : frictional coefficient

$\tilde{\zeta}(x,t)$: water level

$\zeta(x)$: complex amplitude of the water level

$\tilde{u}(x,t)$: $x$-direction current velocity





$u(x)$: complex amplitude of the $x$-direction current velocity

$k$: wavenumber

$H$: water depth

$L$: length of the channel

$m = 1,2$: different channel segments

The governing equation for tidal motion has the form

$$\begin{cases} \dfrac{\partial \tilde{u}_m}{\partial t} = -g \dfrac{\partial \zeta_m}{\partial x} - \gamma_m \tilde{u}_m \\[2mm] \dfrac{\partial \zeta_m}{\partial t} = -H_m \dfrac{\partial \tilde{u}_m}{\partial x} \end{cases}, \tag{A1}$$

where $\tilde{u}_m(x,t)$ and $\tilde{\zeta}_m(x,t)$ can be expressed in the following forms:

$$\tilde{u}_m(x,t) = Re\left(u_m(x)e^{-i\omega t}\right), \tag{A2}$$

$\zeta_m(x,t) = Re(\zeta_m(x)e^{-i\omega t}). \tag{A3}$

Eq. (A1) can be simplified as follows:

$$\begin{cases} -i\omega u_m = -g \dfrac{d\zeta_m}{dx} - \gamma_m u_m \\[2mm] -i\omega \zeta_m = -H_m \dfrac{du_m}{dx} \end{cases}. \tag{A4}$$

From Eq. (A4), we have

$$-i\omega \frac{du_m}{dx} = -g \frac{d^2\zeta_m}{dx^2} - \gamma_m \frac{du_m}{dx} . \tag{A5}$$

Therefore, equivalently,

$$\frac{du_m}{dx} = g \frac{d^2\zeta_m}{dx^2}\left(i\omega - \gamma_m\right)^{-1}. \tag{A6}$$

The second equation in Eq. (4) yields:

$$\frac{du_m}{dx} = i\omega \zeta_m H_m^{-1}. \tag{A7}$$

From Eqs. (A6) and (A7), we obtain

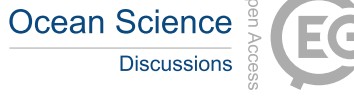



$$\frac{d^2\zeta_m}{dx^2} = i\omega\zeta_m \frac{i\omega - \gamma_m}{gH_m} = \frac{-\omega^2 - i\omega\gamma_m}{gH_m}\zeta_m .$$ (A8)

The above equation can be rewritten in the form

$$\frac{d^2\zeta_m}{dx^2} + \beta_m^{\,2}\zeta_m = 0,$$ (A9)

where

$$\beta_m^{\,2} = k_m^{\,2}(1 + i\mu_m)$$ (A10)

In which $\mu_m = \dfrac{\gamma_m}{\omega}$ is a friction parameter; and $k_m = \dfrac{\omega}{C_m}$ is the wavenumber with $C_m = \sqrt{gH_m}$ representing the wave

velocity.

The solution to Eq. (A9) is:

$$\zeta_m(x) = A_m e^{i\beta_m x} + B_m e^{-i\beta_m x},$$ (A11)

where $\beta_m$ can be expressed in the form

$$\beta_m = \beta_m^{'} + i\beta_m^{''} \ (\beta_m^{'}, \beta_m^{''} \text{ are real numbers}).$$ (A12)

We can obtain the following equations from Eqs. (A10) and (A12):

$$\beta_m^{'2} - \beta_m^{''2} = k_m^{\,2} \Rightarrow \beta_m^{''2} = \beta_m^{'2} - k_m^{\,2},$$ (A13)

and

$$2\beta_m^{'}\beta_m^{''} = \mu_m k_m^{\,2} \Rightarrow \beta_m^{'2}\beta_m^{''2} = \frac{\mu_m^{\,2}k_m^{\,4}}{4},$$ (A14)

or

$$\beta_m^{'2}\left(\beta_m^{'2} - k_m^{\,2}\right) = \frac{\mu_m^{\,2}k_m^{\,4}}{4}, \text{ and}$$ (A15)

$$\beta_m^{'4} - k_m^{\,2}\beta_m^{'2} - \frac{\mu_m^{\,2}k_m^{\,4}}{4} = 0 .$$ (A16)





Therefore,

$$\beta_m^{'2} = \frac{k_m^2 \pm \sqrt{k_m^4 + \mu_m^2 k_m^4}}{2} = \frac{k_m^2\left(1 \pm \sqrt{1+\mu_m^2}\right)}{2}. \tag{A17}$$

Because $\beta_m^{'}$ is any real number, $\beta_m^{'2} \geq 0$, thus,

$$\beta_m^{'2} = \frac{k_m^2(1+\sqrt{1+\mu_m^2})}{2}, \text{ and} \tag{A18}$$

$$\beta_m^{"2} = \frac{k_m^2(-1+\sqrt{1+\mu_m^2})}{2}. \tag{A19}$$

Because Eq. (A11) includes either positive and negative terms for $\beta_m$, we can choose $\beta_m^{'}$ as a positive number. Furthermore, because $\beta_m^{'}$ and $\beta_m^{"}$ have identical signs according to Eq. (A14), they can be written in the forms:

$$\beta_m^{'} = k_m \left(\frac{1+\sqrt{1+\mu_m^2}}{2}\right)^{\frac{1}{2}}, \text{ and} \tag{A20}$$

$$\beta_m^{"} = k_m \left(\frac{-1+\sqrt{1+\mu_m^2}}{2}\right)^{\frac{1}{2}}. \tag{A21}$$

We can obtain the following solution of $u_m(x)$ from Eqs. (A4) and (A11):

$$u_m(x) = \sqrt{\frac{g}{H_m}}(1+i\mu_m)^{-\frac{1}{2}}\left(A_m e^{i\beta_m x} - B_m e^{-i\beta_m x}\right). \tag{A22}$$

The open boundary and matching conditions of the two channels are as follows:

$$\zeta_1(L_1) = a_0, \tag{A23}$$

$$u_2(-L_2) = 0, \tag{A24}$$

$$\zeta_1(0) = \zeta_2(0), \text{ and} \tag{A25}$$

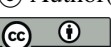



$$H_1 u_1\left(0\right)=H_2 u_2\left(0\right).$$ (A26)

From Eqs. (A11) and (A22–A26), we obtain

$$A_1 e^{i\beta_1 L_1}+B_1 e^{-i\beta_1 L_1}=a_0,$$ (A27)

$$A_2=B_2 e^{2i\beta_2 L_2},$$ (A28)

$$A_1+B_1=A_2+B_2,\text{ and}$$ (A29)

$$A_1-B_1=r\,{p_1}\big/{p_2}\left(A_2-B_2\right),$$ (A30)

where

$$p_1=\left(1+i\mu_1\right)^{1/2},p_2=\left(1+i\mu_2\right)^{1/2},r=\sqrt{\frac{H_2}{H_1}}\,.$$ (A31)

The Eqs. (A27-A30) can be used to determine coefficients $A_1, A_2, B_1, B_2$ through the following process. First, from (A29) and (A30), we obtain

$$A_1=\frac{1+r\,{p_1}\big/{p_2}}{2}A_2+\frac{1-r\,{p_1}\big/{p_2}}{2}B_2,\text{ and}$$ (A32)

$$B_1=\frac{1-r\,{p_1}\big/{p_2}}{2}A_2+\frac{1+r\,{p_1}\big/{p_2}}{2}B_2.$$ (A33)

Therefore, Eq. (A27) can be converted to the following equations.

$$\left(\frac{1+r\,{p_1}\big/{p_2}}{2}A_2+\frac{1-r\,{p_1}\big/{p_2}}{2}B_2\right)e^{i\beta_1 L_1}+\left(\frac{1-r\,{p_1}\big/{p_2}}{2}A_2+\frac{1+r\,{p_1}\big/{p_2}}{2}B_2\right)e^{-i\beta_1 L_1}=a_0$$ (A34)

or





$$\frac{A_2 e^{i\beta_1 L_1}}{2} + \frac{r\,{}^{p_1}\!/_{p_2}}{2} A_2 e^{i\beta_1 L_1} + \frac{B_2 e^{i\beta_1 L_1}}{2} - \frac{r\,{}^{p_1}\!/_{p_2}}{2} B_2 e^{i\beta_1 L_1} +$$

$$\frac{A_2 e^{-i\beta_1 L_1}}{2} - \frac{r\,{}^{p_1}\!/_{p_2}}{2} A_2 e^{-i\beta_1 L_1} + \frac{B_2 e^{-i\beta_1 L_1}}{2} + \frac{r\,{}^{p_1}\!/_{p_2}}{2} B_2 e^{-i\beta_1 L_1} = a_0$$

(A35)

This results further gives

$$A_2 \cos \beta_1 L_1 + ir\,{}^{p_1}\!/_{p_2}\, A_2 \sin \beta_1 L_1 + B_2 \cos \beta_1 L_1 - ir\,{}^{p_1}\!/_{p_2}\, B_2 \sin \beta_1 L_1 = a_0 ,$$

(A36)

or

$$\left(A_2 + B_2\right)\cos \beta_1 L_1 + ir\,{}^{p_1}\!/_{p_2}\, \left(A_2 - B_2\right)\sin \beta_1 L_1 = a_0 .$$

(A37)

Substituting Eq. (A28) for $A_2$ in Eq. (A37), we obtain:

$$\left(1 + e^{2i\beta_2 L_2}\right)B_2 \cos \beta_1 L_1 + ir\,{}^{p_1}\!/_{p_2}\, \left(e^{2i\beta_2 L_2} - 1\right)B_2 \sin \beta_1 L_1 = a_0 .$$

(A38)

Thus,

$$
\begin{aligned}
B_2 &= \frac{a_0}{\left(1 + e^{2i\beta_2 L_2}\right)\cos \beta_1 L_1 + ir\,{}^{p_1}\!/_{p_2}\, \sin \beta_1 L_1 \left(e^{2i\beta_2 L_2} - 1\right)} \\[2mm]
&= \frac{a_0}{\dfrac{e^{-i\beta_2 L_2} + e^{i\beta_2 L_2}}{e^{-i\beta_2 L_2}} \cos \beta_1 L_1 + ir\,{}^{p_1}\!/_{p_2}\, \sin \beta_1 L_1 \dfrac{-e^{-i\beta_2 L_2} + e^{i\beta_2 L_2}}{e^{-i\beta_2 L_2}}} , \\[2mm]
&= \frac{a_0 e^{-i\beta_2 L_2}}{2\cos \beta_2 L_2 \cos \beta_1 L_1 - 2r\,{}^{p_1}\!/_{p_2}\, \sin \beta_1 L_1 \sin \beta_2 L_2} \\[2mm]
&= \frac{a_0}{2} R_0 e^{-i\beta_2 L_2}
\end{aligned}
$$

(A39)

where





$$R_0 = \frac{1}{\cos\beta_2 L_2 \cos\beta_1 L_1 - r\,{}^{p_1}\!/_{p_2}\,\sin\beta_1 L_1 \sin\beta_2 L_2} \,. \tag{A40}$$

We can then obtain the value of $A_2$ through Eq. (A28):


$$A_2 = \frac{a_0}{2} e^{i\beta_2 L_2} \frac{1}{\cos\beta_2 L_2 \cos\beta_1 L_1 - r\,{}^{p_1}\!/_{p_2}\,\sin\beta_1 L_1 \sin\beta_2 L_2} = \frac{a_0}{2} R_0 e^{i\beta_2 L_2} \,. \tag{A41}$$

Substituting Eqs. (A39) and (A40) into Eq. (A11) gives

$$\begin{aligned}\zeta_2(x) &= \frac{a_0}{2} R_0 e^{i\beta_2 L_2} e^{i\beta_2 x} + \frac{a_0}{2} R_0 e^{-i\beta_2 L_2} e^{-i\beta_2 x} \\ &= a_0 R_0 \cos\beta_2(x + L_2)\end{aligned} \,. \tag{A42}$$

We can then obtain the expressions of the coefficients of $\zeta_1(x)$ from Eqs. (A32), (A33), (A39) and (A40) as follows:

$$A_1 = \frac{1 + r\,{}^{p_1}\!/_{p_2}}{2} \frac{a_0}{2} R_0 e^{i\beta_2 L_2} + \frac{1 - r\,{}^{p_1}\!/_{p_2}}{2} \frac{a_0}{2} R_0 e^{-i\beta_2 L_2} \,, \tag{A43}$$


$$B_1 = \frac{1 - r\,{}^{p_1}\!/_{p_2}}{2} \frac{a_0}{2} R_0 e^{i\beta_2 L_2} + \frac{1 + r\,{}^{p_1}\!/_{p_2}}{2} R_0 e^{-i\beta_2 L_2} \,. \tag{A44}$$

Thus,



$$\zeta_1(x) = (\frac{1+r\,{p_1}/{p_2}}{2}\frac{a_0}{2}R_0 e^{i\beta_2 L_2} + \frac{1-r\,{p_1}/{p_2}}{2}\frac{a_0}{2}R_0 e^{-i\beta_2 L_2})e^{i\beta_1 x}$$

$$+ (\frac{1-r\,{p_1}/{p_2}}{2}\frac{a_0}{2}R_0 e^{i\beta_2 L_2} + \frac{1+r\,{p_1}/{p_2}}{2}\frac{a_0}{2}R_0 e^{-i\beta_2 L_2})e^{-i\beta_1 x}$$

$$= \frac{a_0}{2}R_0 \left[ \left( \frac{e^{i\beta_2 L_2}e^{i\beta_1 x}}{2} + \frac{e^{-i\beta_2 L_2}e^{i\beta_1 x}}{2} + \frac{e^{i\beta_2 L_2}e^{-i\beta_1 x}}{2} + \frac{e^{-i\beta_2 L_2}e^{-i\beta_1 x}}{2} \right) \\ + r\,{p_1}/{p_2} \left( \frac{e^{i\beta_2 L_2}e^{i\beta_1 x}}{2} - \frac{e^{-i\beta_2 L_2}e^{i\beta_1 x}}{2} - \frac{e^{i\beta_2 L_2}e^{-i\beta_1 x}}{2} + \frac{e^{-i\beta_2 L_2}e^{-i\beta_1 x}}{2} \right) \right]. \tag{A45}$$

$$= a_0 R_0 \left[ \cos(\beta_2 L)\cos(\beta_1 x) - r\frac{p_1}{p_2}\sin(\beta_2 L)\sin(\beta_1 x) \right]$$

## Competing interests

The authors declare that they have no conflicts of interest.

## Acknowledgements

This study was supported by the National Key Research and Development Program of China (2017YFC1404200), the NSFC-Shandong Joint Fund for Marine Science Research Centers (Grant No. U1406404), and the Basic Scientific Fund for National Public Research Institutes of China (Grant No. 2015G02).

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



**Figure captions**

**Figure 1: Study area. The contours show the water depth distribution in meters. The red line B is the mouth cross section of the Gulf of Thailand (GOT).**

**Figure 2: Response functions (amplitude gains as a function of frequency) of the South China Sea body (SCSB) (a) and the GOT (b) for different bottom topographies of the SCSB.**

**Figure 3: Comparison of the response functions of the SCSB with and without the GOT.**

**Figure 4: The two-channel model configuration representing the SCSB-GOT system.**

**Figure 5: Response function at the locations $x = -L_2$ (a) and $x = 0$ (b) with and without friction.**

**Figure 6: The amplitude gain as a function of $x$ for different peak frequencies with friction (a) and without friction (b).**




**Tables**

**Table 1. Resonant frequencies and periods obtained from the four experiments.**

| Exp | Topography | | Resonant frequency /d$^{-1}$ | | Resonant period /h | |
|---|---|---|---|---|---|---|
| | SCSB | GOT | SCSB | GOT | SCSB | GOT |
| 1 | Real | Real | 0.99 | 1.01 | 24.36 | 23.79 |
| 2 | Half | Real | 0.75 | 0.75 | 32.00 | 32.00 |
| 3 | Twice | Real | 1.49 | 1.50 | 16.10 | 16.00 |





**Table 2. The resonant frequencies and corresponding periods at $x = -L_2$ and $x = 0$.**

| Solution | $x = -L_2$ | | | | | | $x = 0$ | | | | | |
|---|---|---|---|---|---|---|---|---|---|---|---|---|
| | Peak frequency /d$^{-1}$ | | | Peak period /h | | | Peak frequency /d$^{-1}$ | | | Peak period /h | | |
| 1 | 0.558 | 1.040 | 1.848 | 43.01 | 23.08 | 12.99 | - | 1.040 | - | - | 23.08 | - |
| 2 | 0.545 | 1.044 | 1.856 | 44.04 | 22.99 | 12.93 | 0.545 | 1.044 | 1.856 | 44.04 | 22.99 | 12.93 |

Notes: Solution 1 and 2 are respectively for the cases in the presence and in the absence of friction



**Figures**

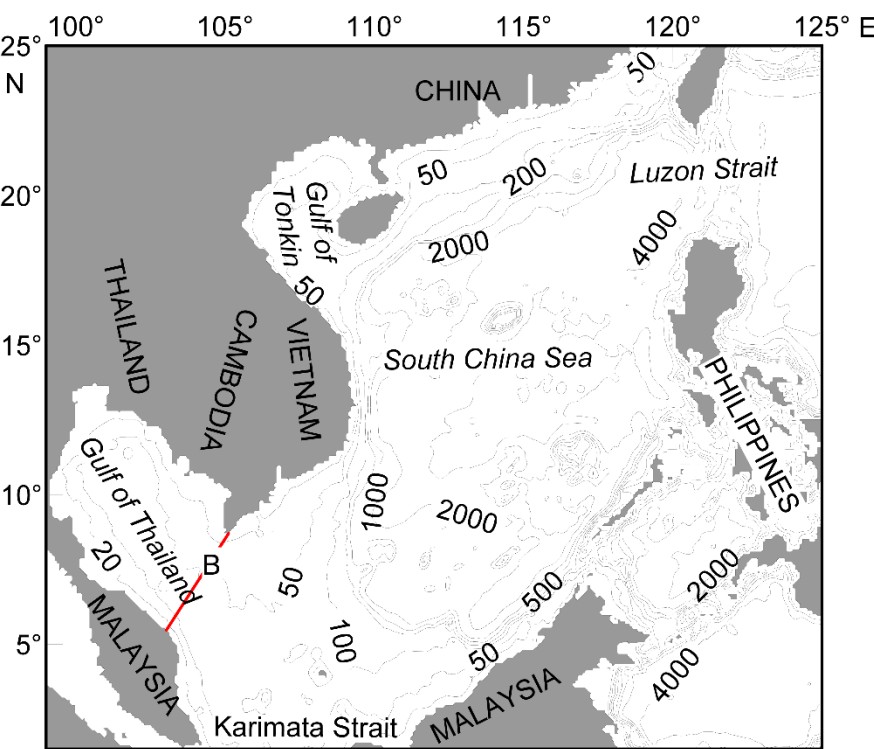

**Figure 1**



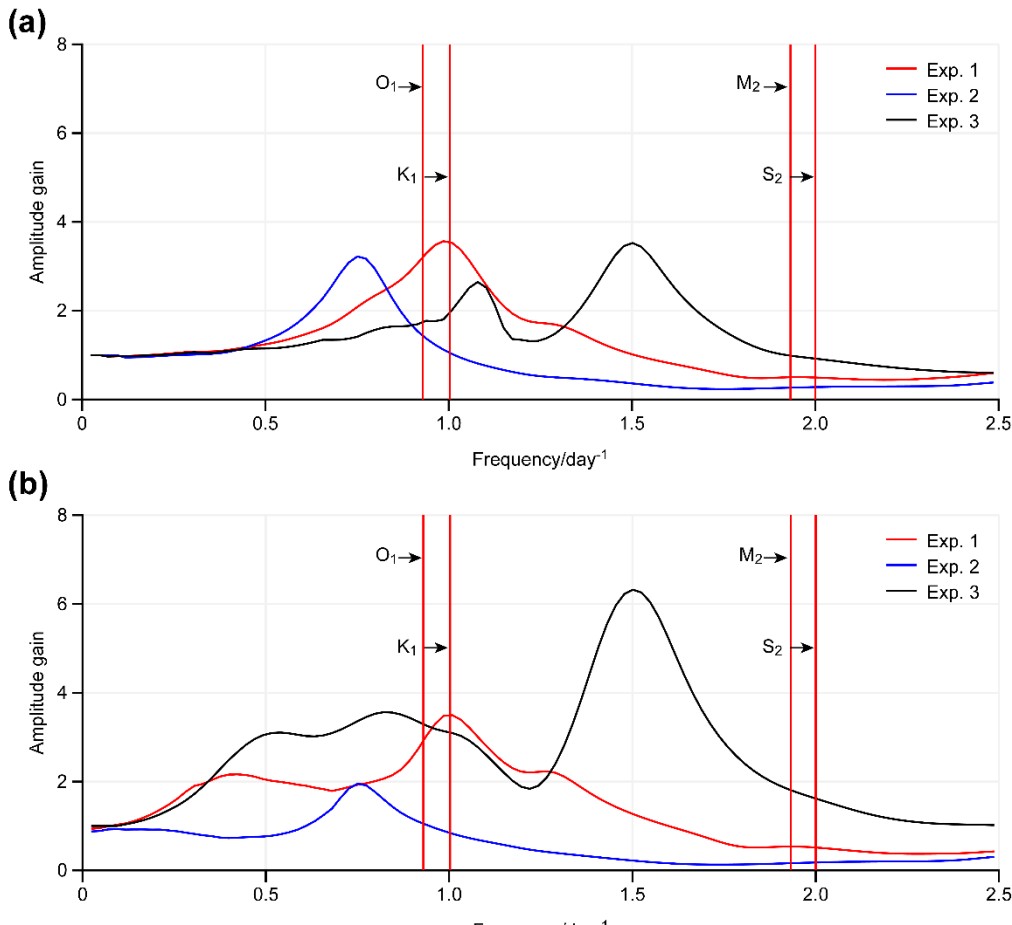

**Figure 2**





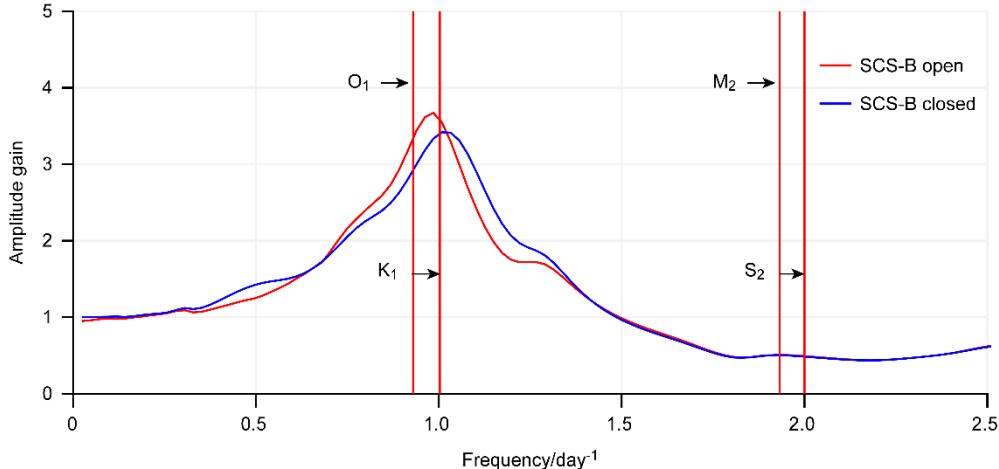

**Figure 3**



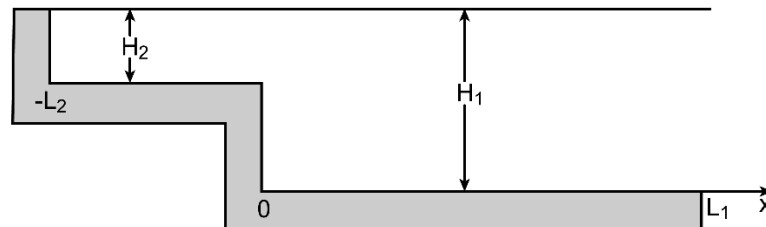


**Figure 4**



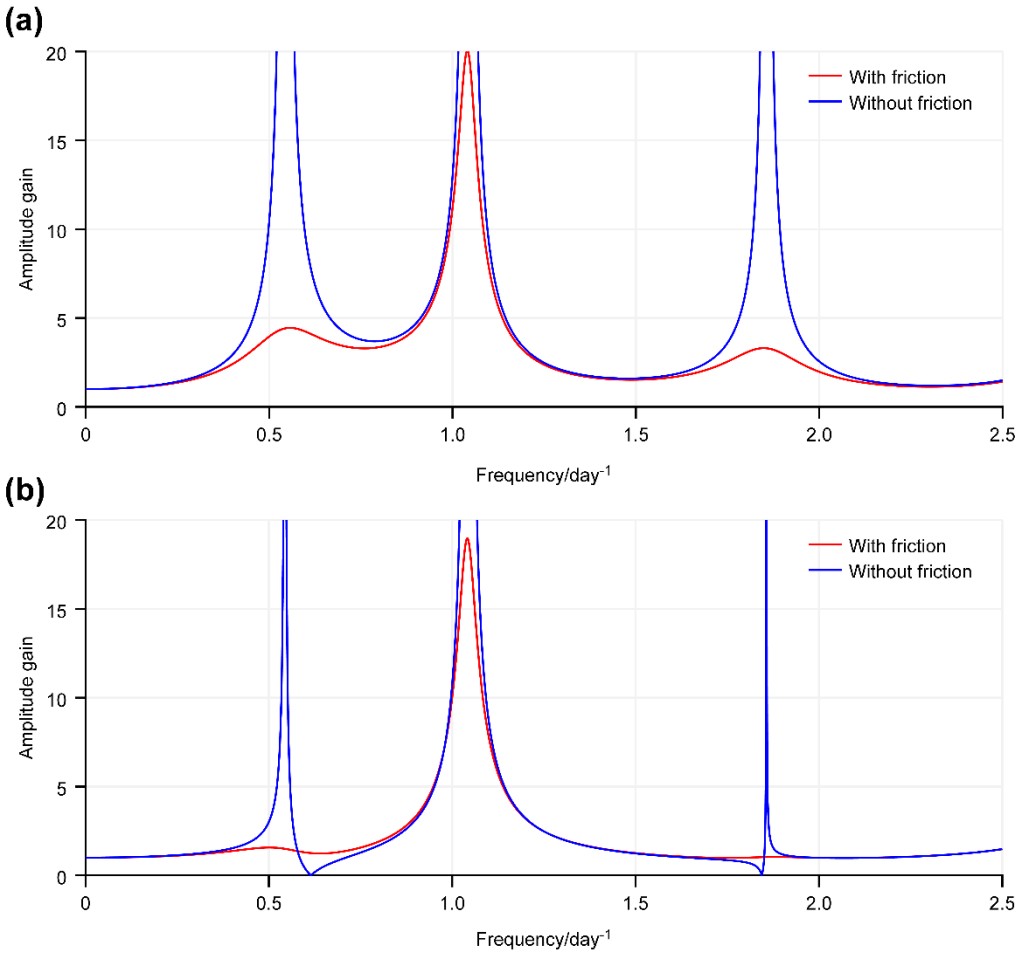

**Figure 5**




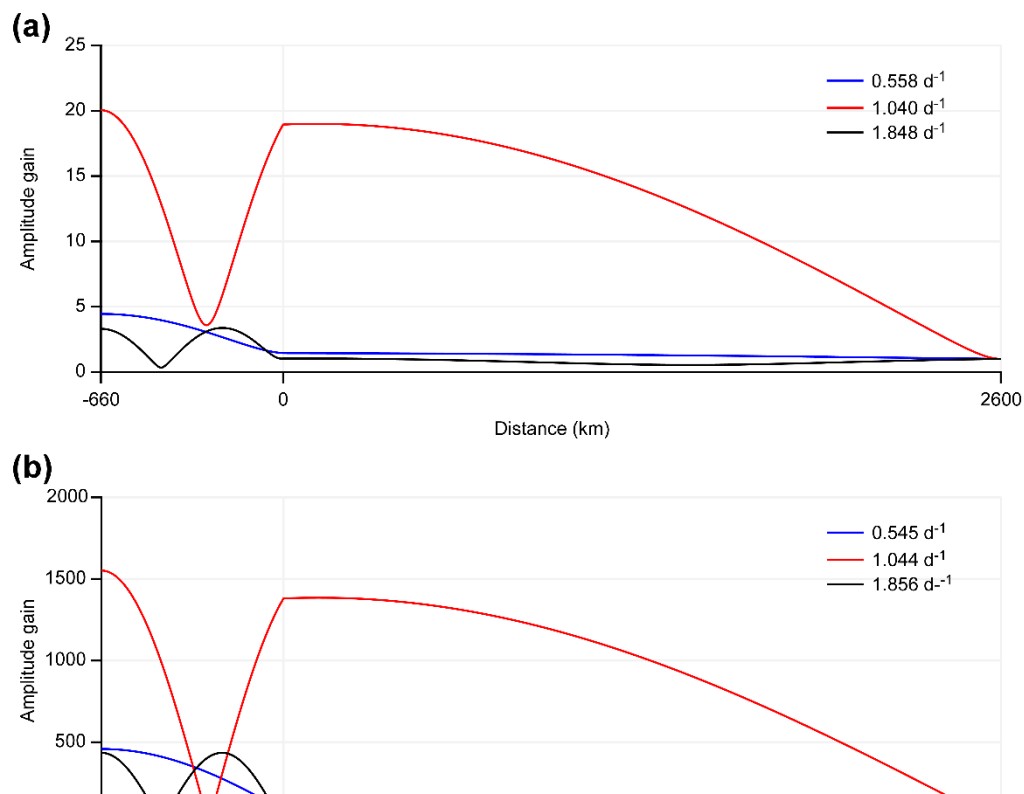

**Figure 6**