# Peer review of "Tidal resonance in the Gulf of Thailand"

_Ocean Science, 2018_

## Referee Comment (RC1) · Anonymous Referee #1 · 24 Sep 2018

General Comments:

Overall, the manuscript investigates the resonant period of the Gulf of Thailand (GOT) via numerical experiments and tries to establish a conceptual understanding of resonance in the gulf over two-channel model. The authors found that the resonant period of the GOT is closely related to that of the South China Sea body (SCSB) and is close to the period of the major diurnal tide, K1. They speculate that the resonance of the SCSB has a critical impact on the resonance of the GOT. On contrary, the resonance of the GOT has little influence on the resonance of the SCSB. I suggest that though this work seems to present interesting results speculating the interconnection of resonance between coastal bays and deep sea. However, the substantial analysis/discussion for convincing their findings/conclusions are inadequate and not up to the standard of the journal. For consideration of OS editor, some critical issues are addressed below:

[Figure]

Specific Comments:

1. Theoretically, the characteristic of the effective region for the resonance of long waves in the semi-enclosed sea can be calculated via the phase speed. For the GOT, the effective length of the basin for resonance of the diurnal tides can be approximately 1700 km. Besides, the co-tidal chart for K1 tide in the gulf suggests more precision length such as 1500 km. From this information, the resonant periods for the fundamental and first mode would be calculated as 73.8 and 24.61 hours (based on semi-enclosed basin formula). From these numbers, we could say that the period of diurnal tides in the gulf can be predominated by influence of the first mode instead of the fundamental mode. The role of the quarter wavelength resonant theory on tidal resonance in the GOT is insignificant and is easy to prove. However, this issue tends to be highlighted in the abstract and conclusion of the manuscript. But, it does not represent a substantial contribution to scientific progress in oceanography.

2. I wonder why the length of the GOT system is limited by the 660 km. The mentioned length may represent only the territorial sea of Thailand but does not involve the effective resonator system for tidal waves in true nature. Contrary, the size of the effective system should be larger as the entire western shelf of the South China Sea or Sunda Shelf (see the previous comment). Therefore, I suggest that the perception of a basin resonance oscillator and the dynamics of tidal waves in the GOT, the division of the computational domain, especially, the judgment of the authors for the application of classic quarter wavelength resonant theory for determining a diurnal resonant period of the GOT are altogether possible misconceptions.

3. There are several resonance mechanisms (standing waves or basin mode and shelf mode) that might control oscillation of sea levels in the GOT system. Entirely, the impact of the standing waves modes associated with the period of approximately 24 hours is mostly accounted for. It is recognized that their modal structure distribution (nodal and anti-nodal bands) along the major axis of the system (the distance from the inner GOT to Kalimata strait, NS mode). Supported by the geometry defined by the

distance from the Malaysia Peninsular to the eastern of the Taiwan Strait, the modal structure of mentioned period may also be fitted into the SCSB. But, it should have a different modal structure (East-West, EW mode). The existence of the mentioned modal structures revealed in the manuscript. Besides, the experiment in determining the effect of bottom topographies of the SCSB on the resonant response of the GOT is presented. As present in this part of the results, it seems that the consistency of the resonant periods are the main reason to judge that the GOT is not an independent sea area regarding tidal resonance. On the other hand, the amplification mechanism is not involved although the response (amplitude gain) of the GOT is probably higher than that of the SCSB (See Exp. 3 Result). The real phenomenon similar to the Exp.3 can be such as the resonance of M2 in the Bay of Bengal, Andaman seas and Malaca Strait.

The part of the deeper and shallow sea of the mentioned system may have the same resonant period but amplification become more intensified near the shelf zone. Indeed, the resonance of the Andaman seas and Malaca Strait are not independent from the Bay of Bengal. But, they have the locality regarding the modal structure and amplification processes. Importantly, we might explain dynamic of tidal waves in the mentioned area as an influence of a combined-role of basin and local resonance modes. I suggest that this concept would also explain the interconnection of the GOT and SCSB. Hence, I reject that judgment as mentioned above because the locality of the GOT and the SCSB are found in their results. Moreover, for the idealized model part, the authors only show preliminary results that mostly identical to the numerical experiment. They do not present some discussion showing the benefit of the model to gain more comprehension of the tidal resonance in the GOT.

---

## Referee Comment (RC2) · D.J. Webb (Referee) · 29 Oct 2018

The oceans and marginal seas around South-east Asia are unusual in that the diurnal tides are often much more significant than in the rest of the world's ocean. In this paper the tides of the Gulf of Thailand are investigated to see how resonances enhance the tides of the region. The main result is that the high diurnal tides are not due to a resonance of the Gulf of Thailand but that they are probably due to a quarter wave resonance of the South China Sea. The paper builds on the model study of Cui et al (2015) but also includes an analytic 1-D model which supports the quarter-wave hypothesis.

The paper is well written and easy to understand and although I have some serious criticisms of the work I would like to commend the authors on the standard of their discussion paper.

[Figure]

Detailed comments

1. Abstract.

After reading the paper and that of Cui et al (2015) it seems obvious that it is the South China Sea which is responsible for the resonance. Thus changing the depth of the South China Sea changes the frequency of the resonance (fig 2) and the analytic resonance around 1 cy/day comes for the cos(beta_1 L_1) term in the equation on line 159. However the abstract says that the resonant period of the Gulf of Thailand is itself close to 1 cycle/day - which is incorrect.

It would be more correct to say that the South China Sea and the surrounding sea together have a resonance around 1 cycle/day which is primarily due to the South China Sea having very close to a quarter wavelength standing wave at this frequency. Although the Gulf of Thailand does have a large amplitude response around 1 cycle per day the results indicate that this is just a passive response of the Gulf to the increased amplitude of the main South China Sea wave along the Gulf's southern boundary.

2. Lines 19, 22, 24

It would help if the geographical features Taiwan Strait, Mindoro Strait, Balabac Strait and any others referred to in the text were were included in figure 1.

3. Line 32

In some parts of the literature there is a tendency to refer to resonances in terms of their period. However because the angular velocity of resonances (or their frequencies) often form an arithmetic series corresponding 1/4, 3/4, 5/4, etc wavelengths I would recommend replacing periods here by angular velocities (or frequencies), possibly with the periods in brackets for those that need them.

4. Line 47-48

It is not that the resonant periods are related but that the two features are parts of the

same resonance, the angular velocity of the resonance being determined primarily by the physical properties of the South China Sea.

5. Line 72: The Numerical Model

It is only in the code availability section that you say that you use the Princeton Ocean Model. I think this needs to be mentioned earlier in the paper as there are many types of ocean model of varying quality.

The Princeton Model is well known and is usually acknowledged to be of good quality. However it includes many options and parameters so, as in any other realistic model study, it is important to show that the version in use can realistically represent the actual tides in the region being studied. For this reason the paper needs an example of the model K1 (and/or O1) tides of the region and comparison with actual tidal observations either in the form of a chart or in the form of comparisons at key tide gauge positions. I realise that Cui et al (2015) did not do this, but if I had refereed their paper I would have made the same point.

You could also do with a figures showing the flux of tidal energy for both the realistic K1 tidal forcing and with a constant amplitude on the boundary as in your test experiments, to show that the main influx of tidal energy is through the Luzon Strait. If you do not do this it is possible that your analytic model which is based on this assumption is not valid.

6. Line 72: Boundary Condition

More information is needed on the boundary of the model. The domain described in the text seems to be similar to Cui et al (2015) which makes me suspect you used the same code in the same configuration. However the other paper shows a southern boundary south of the Equator, whereas according to your text the present one is north of the Equator. Why the difference?

Figure 1, which the caption calls the study area, shows only part of the model domain.

Why is this? When I first read the introduction and saw this figure, I assumed that this included the model domain with say Luzon Strait as an open boundary and the regions you described as having negligible fluxes as closed boundaries. Your need to make the difference clearer earlier in the paper.

You say that the northern and eastern boundaries as set well away from the South China Sea to limit the effect of the (fixed) boundary condition on the resonances of the South China Sea. However what about the boundaries to the south and east?

6. Line 64 and 80-90: Real and complex variables

Analysis of waves and oscillating systems tends to be a lot neater and easier to understand when the physical wave is treated as the real part of a function of the type $A(x)$ $\exp(-i \omega t)$ where A is a complex number and 'i' is the complex i. Then your G and phi are just the amplitude and phase of a complex response function. The appendix would also be a lot shorter if you used complex variables whenever possible.

7. Lines 94-96

This is a bit of a mess and needs to be rewritten. From Cui at al. (2015) you know that there are resonant like features which affect both the Gulf of Thailand and the South China Sea. We know that if the Gulf of Thailand was removed, changing the depth of the South China Sea would affect its resonances. Thus what you are really doing here is to see how much changing these resonances affects the resonances of the combined system. (You could have also carried out runs with changed depths in the Gulf of Thailand - in fact I am surprised that you didn't).

What you are not doing here is finding out how the resonances of the South China Sea are affecting (i.e. changing the shape and frequency of) the localised resonances of the Gulf of Thailand.

8. Line 100- : The Results

The paper does not specify the geographical location used for figures 2a and 2b.

The three SCSB peaks referred to in table 1 refer to the three main peaks of fig 2a. Changing the depth by a factor of 2 seems to change the frequencies by roughly sqrt(2), but this is not discussed.

Expt 3 shows a second resonance near 1 cycle/day which also seems to have an effect in fig 2b. Changing the depth of the South China Sea will change the frequency of resonances but will not generate new ones. So what is this feature of the South China Sea affecting the Gulf of Thailand?

In the case of fig 2b, representing the Gulf of Thailand response, there are indeed three main peaks matching the peaks in the South China sea plot - but there is also a lot else going on, especially in expt 1 and 2. What resonances are these?

It should help if the paper illustrated the amplitude and and phase of key resonances. The simplest solution would be to give the amplitude and phase of the solution when forced at the frequencies of the peaks in the response function. A better alternative would be to fit the (complex) response function $R(x,w)$ at a set of w's around each of the resonance peaks to the equation $A(x)/(w-w0) + B(x) + C(x)*(w-w0)$ Where A, B, C, w0 are complex, x is position, w is the (real) angular velocity of the forcing and w0 the estimated (complex) angular velocity of the resonance. $A(x)$ would then be a better approximation to the amplitude and phase of the true resonance.

Following the changes, the conclusions to this section need to be rewritten.

9. Line 128

Semantics - the theory is 'applicable' to the Gulf of Thailand but does not explain the enhanced tides around 1 cycle day (although it might explain an enhancement if forced at 2 cycles/day).

10. Lines 147-160

It would be best if most of this was kept in the appendix. All you really need is eqn 15 and the approximation when $r*p\_1/p\_2$ is small.
11. Lines 165-185.

I think this needs a little more thought. You should be able to show that the resonances near 0.5 and 2 cycles per day are resonances of the short channel (where cos(beta_2 L_2) is zero) and the one near 1 cycle per day is a resonance of the long channel (where cos(beta_1 L_1) is zero). Then friction reduces the amplitude of the shallow short channel resonances but has little effect on the amplitude of the long channel resonances except within the shallow channel.

12. Lines 189-195

I suppose my main point here is that the diurnal resonance affecting the Gulf of Thailand in not 'closely related' to that affecting the South China Sea. Instead it is exactly the same resonance.

13. Appendix

This seems a bit long for the content. I suggest that you cut it down in size, trying to make it more elegant and leaving out some of the obvious steps.

14. !!

I know it is bad form but I was asked to review this paper because I have worked on tidal resonances. The authors may not be aware of the following papers but they may provide a more understandable background to some of my comments above: https://nora.nerc.ac.uk/id/eprint/271197 https://www.ocean-sci.net/10/411/2014/

Regards,

David Webb.

---

## Author Comment (AC1) · 9 Jan 2019

Dear Referee and EditorïijŽ Reply: We sincerely thank the referee for his careful reading of our manuscript and comments. We have revised this paper and addressed all these comments; our responses are given below. In this response, the referee's comments are copied in black, our replies are shown in red, and the following abbreviations are used: R1 – Revision #1 - an updated manuscript, which will be submitted as a supplement to this response.

Please also note the supplement to this comment:
https://www.ocean-sci-discuss.net/os-2018-97/os-2018-97-AC1-supplement.zip
* * *

---

## Author Comment (AC2) · 9 Jan 2019

Dear Dr. Webb we sincerely thank you for your careful reading of our manuscript and your constructive comments and suggestions, which are of great help in improving our study. We have addressed all these comments; our responses are given below. In this response, your comments are copied in black, our replies are shown in red, and the following abbreviations are used: R1 – Revision #1 - an updated manuscript, which will be submitted as a supplement to this response.

Please also note the supplement to this comment:
https://www.ocean-sci-discuss.net/os-2018-97/os-2018-97-AC2-supplement.zip

---

## Author Response (AR1)

General Comments:

Overall, the manuscript investigates the resonant period of the Gulf of Thailand (GOT) via numerical experiments and tries to establish a conceptual understanding of resonance in the gulf over two-channel model. The authors found that the resonant period of the GOT is closely related to that of the South China Sea body (SCSB) and is close to the period of the major diurnal tide, K1. They speculate that the resonance of the SCSB has a critical impact on the resonance of the GOT. On contrary, the resonance of the GOT has little influence on the resonance of the SCSB. I suggest that though this work seems to present interesting results speculating the interconnection of resonance between coastal bays and deep sea. However, the substantial analysis/discussion for convincing their findings/conclusions are inadequate and not up to the standard of the journal. For consideration of OS editor, some critical issues are addressed below:

Reply: We sincerely thank the referee for his careful reading of our manuscript and comments. We have revised this paper and addressed all these comments; our responses are given below.

In this response, the referee's comments are copied in black, our replies are shown in red, and the following abbreviations are used:

R1 – Revision #1 - an updated manuscript, which will be submitted as a supplement to this response.

Specific Comments:

1. Theoretically, the characteristic of the effective region for the resonance of long waves in the semi-enclosed sea can be calculated via the phase speed. For the GOT, the effective length of the basin for resonance of the diurnal tides can be approximately 1700 km. Besides, the co-tidal chart for K1 tide in the gulf suggests more precision length such as 1500 km. From this information, the resonant periods for the fundamental and first mode would be calculated as 73.8 and 24.61 hours (based on semi-enclosed basin formula). From these numbers, we could say that the period of diurnal tides in the gulf can be predominated by influence of the first mode instead of the fundamental mode. The role of the quarter wavelength resonant theory on tidal resonance in the GOT is insignificant and is easy to prove. However, this issue tends to be highlighted in the abstract and conclusion of the manuscript. But, it does not represent a substantial contribution to scientific progress in oceanography.

Reply: The GOT is a subsidiary gulf of the SCS, and the SCS is mainly composed of deep-sea basin and continental shelf (Figure 1). The SCS deep basin (abbreviated as SCSDB, where the water depth is more than 500 m, the blue line in Figure 1) connects with the GOT through the Sunda shelf (abbreviated as SS, where the water depth is less than 500 m). The average depth of the SCSDB is 2500 m, and the average depth of the SS and GOT are 66 m and 36 m, respectively. The water depth from the SCSDB to the SS varies dramatically (Figure 1).

The average depth within 1700 km (the length of the red line in Figure 1) of the semi-enclosed sea area is 173 m, of which the average depths of the GOT and SS are respectively 36 m and 66 m, and the average depth of a small part of the SCSDB is approximately 1000 m. According to the semi-closed basin formula, the resonance period should be approximately 15 hours, not 24 hours. If the range of 1700 or 1500 km is considered covering the whole continental shelf (including GOT and SS), its average depth is only 46 m and its resonance period is between 26 and 29 hours, not 24 hours.

Furthermore, in Cui et al. (2015), we find that the resonant period of the SCSDB is approximately one

day because the one-quarter wavelength resonance in the SCSB. In this paper, we carried out two experiments by changing the depth of SCSB; the depths are one half and two times that of the real depth, and we find that the resonant frequencies (periods) are approximately 0.5 cycle/day (period is 43 hours) and 1.5 cycle/day (period is 16 hours), respectively. The experimental results are consistent with those from the one-quarter wavelength of the SCSB. Again, it is proved that the strong diurnal tide in the SCSB is caused by the quarter-wavelength resonance, so the SCSB (covering the SS and SCSDB) can be regarded as a system subject to the one-quarter wavelength resonance.

Both the GOT and the Gulf of Tonkin are subsidiary gulfs of the SCS, and the strong diurnal tide in the Gulf of Tonkin is caused by the quarter-wavelength resonance (Cui et al, 2015; Fang et al, 1999). We speculate whether the strong diurnal tide in the GOT can be explained by the quarter-wavelength resonance, but we find that it cannot. We don't know the reasons for the strong response around one cycle/day in the GOT. Since the tidal energy of the GOT comes mainly from the SCS, we speculate that the strong response in the GOT may be related to the SCS. In R1, we conclude that the GOT does have a large amplitude response around one cycle/day, and the results indicate that is just a passive response of the gulf to the increased amplitude of the SCSB along its southern boundary.

[Figure]

Figure 1. The length of the red line 1 is 1700 km; the blue line shows the 500 m isobath.

2. I wonder why the length of the GOT system is limited by the 660 km. The mentioned length may represent only the territorial sea of Thailand but does not involve the effective resonator system for tidal waves in true nature. Contrary, the size of the effective system should be larger as the entire western shelf of the South China Sea or Sunda Shelf (see the previous comment). Therefore, I suggest that the perception of a basin resonance oscillator and the dynamics of tidal waves in the GOT, the division of the computational domain, especially, the judgment of the authors for the application of classic quarter wavelength resonant theory for determining a diurnal resonant period of the GOT are altogether possible misconceptions.

Reply: Please kindly see the previous reply.

3. There are several resonance mechanisms (standing waves or basin mode and shelf mode) that might control oscillation of sea levels in the GOT system. Entirely, the impact of the standing waves modes associated with the period of approximately 24 hours is mostly accounted for. It is recognized that their modal structure distribution (nodal and anti-nodal bands) along the major axis of the system (the distance from the inner GOT to Kalimata strait, NS mode).

Reply: The SCSB plays a decisive role in the response peak in the GOT at the frequency of one cycle/day, and one-quarter wavelength of this frequency is approximately equal to the length of the SCSB, which leads a significant peak at the south of the SCSB, where is also the entrance to the GOT. The significant amplitude at the entrance of the GOT causes the GOT to have a strong response at the frequency of one cycle/day. The large amplitude in the GOT at this frequency is just a passive response to the increased amplitude caused by the one-quarter wavelength resonance in the SCSB.

Supported by the geometry defined by the distance from the Malaysia Peninsular to the eastern of the Taiwan Strait, the modal structure of mentioned period may also be fitted into the SCSB. But, it should have a different modal structure (East-West, EW mode). The existence of the mentioned modal structures revealed in the manuscript.

Reply: Please kindly see the previous reply.

Besides, the experiment in determining the effect of bottom topographies of the SCSB on the resonant response of the GOT is presented. As present in this part of the results, it seems that the consistency of the resonant periods are the main reason to judge that the GOT is not an independent sea area regarding tidal resonance. On the other hand, the amplification mechanism is not involved although the response (amplitude gain) of the GOT is probably higher than that of the SCSB (See Exp. 3 Result). The real phenomenon similar to the Exp.3 can be such as the resonance of M2 in the Bay of Bengal, Andaman seas and Malaca Strait.

Reply: Please kindly see the previous reply.

The part of the deeper and shallow sea of the mentioned system may have the same resonant period but amplification become more intensified near the shelf zone. Indeed, the resonance of the Andaman seas and Malaca Strait are not independent from the Bay of Bengal. But, they have the locality regarding the modal structure and amplification processes. Importantly, we might explain dynamic of tidal waves in the mentioned area as an influence of a combined-role of basin and local resonance modes. I suggest that this concept would also explain the interconnection of the GOT and SCSB. Hence, I reject that judgment as mentioned above because the locality of the GOT and the SCSB are found in their results.

Reply: Please kindly see the previous reply.

Moreover, for the idealized model part, the authors only show preliminary results that mostly identical to the numerical experiment. They do not present some discussion showing the benefit of the model to gain more comprehension of the tidal resonance in the GOT.

Reply: We added a new part containing content about this idealized model in R1 as follows: "If we apply the quarter-wavelength resonance theory to channel 1, we can obtain resonant frequencies of 0.99 $d^{-1}$. If we apply the quarter-wavelength and three-quarter-wavelength resonance theories to channel 2 we can obtain resonant frequencies of 0.61 and 1.84 $d^{-1}$, respectively. Therefore, we can conclude that the major peaks around the frequency of 1.04 $d^{-1}$ in Figure 6 are caused by resonance in channel 1. This indicates that channel 1 plays a determinative role in the two-channel system. Similarly, we can also conclude that the secondary and third peaks around the frequencies of 0.55 and 1.85 $d^{-1}$ in Figure 6 are caused by resonances in channel 2, associated with the quarter-wavelength and three-quarter-wavelength

resonances. Although the frequencies of the peaks shown in Figure 6 correspond well with those estimated based on the quarter-wavelength and three-quarter-wavelength theories, there are small discrepancies. This is due to the connection of the two channels. In fact, the resonant frequencies of the two-channel system also depend on the depth ratio of two channels, as shown in Eq. (14). In comparison to channel 2, the secondary, especially the third peak, in channel 1 is much more less significant. This can be explained as follows: The tidal incident wave from the channel 1 partially enters channel 2 across the steep topography at $x = 0$, and here, the rest of the wave is reflected. The reflected wave is superimposed with the incident wave, and tidal resonance occurs around the frequency of 1.04 d$^{-1}$. That is, the steep topography at $x = 0$ acts as a wall for channel 1, which causes the quarter-wavelength resonance to occur in the channel. Furthermore, the steep topography can also block most energy of the wave in channel 2 from entering channel 1. Therefore, the relatively large amplitudes in channel 2 at frequencies around 0.55 d$^{-1}$ and 1.85 d$^{-1}$ are not obvious in channel 1 under the action of friction".

Response to Dr D.J. Webb (Referee)

The oceans and marginal seas around South-east Asia are unusual in that the diurnal tides are often much more significant than in the rest of the world's ocean. In this paper the tides of the Gulf of Thailand are investigated to see how resonances enhance the tides of the region. The main result is that the high diurnal tides are not due to a resonance of the Gulf of Thailand but that they are probably due to a quarter wave resonance of the South China Sea. The paper builds on the model study of Cui et al (2015) but also includes an analytic 1-D model which supports the quarter-wave hypothesis.

The paper is well written and easy to understand and although I have some serious criticisms of the work I would like to commend the authors on the standard of their discussion paper.

Reply: Dear Dr. Webb, we sincerely thank you for your careful reading of our manuscript and your constructive comments and suggestions, which are of great help in improving our study. We have addressed all these comments; our responses are given below.

In this response, your comments are copied in black, our replies are shown in red, and the following abbreviations are used:

R1 – Revision #1 - an updated manuscript, which will be submitted as a supplement to this response.

1. Abstract.

After reading the paper and that of Cui et al (2015) it seems obvious that it is the South China Sea which is responsible for the resonance. Thus changing the depth of the South China Sea changes the frequency of the resonance (fig 2) and the analytic resonance around 1 cy/day comes for the cos(beta_1 L_1) term in the equation on line 159. However, the abstract says that the resonant period of the Gulf of Thailand is itself close to 1 cycle/day - which is incorrect.

It would be more correct to say that the South China Sea and the surrounding sea together have a resonance around 1 cycle/day which is primarily due to the South China Sea having very close to a quarter wavelength standing wave at this frequency. Although the Gulf of Thailand does have a large amplitude response around 1 cycle per day the results indicate that this is just a passive response of the Gulf to the increased amplitude of the main South China Sea wave along the Gulf's southern boundary.

Reply: We agree with this comment and have added the following sentence in the abstract: "We find that the resonant frequency around 1 cycle per day in the main area of the South China Sea can be explained with the quarter-wavelength theory, and the large-amplitude response at this frequency in the Gulf of Thailand is basically a passive response of the gulf to the increased amplitude of the wave in the southern portion of the main area of the South China Sea".

2. Lines 19, 22, 24

It would help if the geographical features Taiwan Strait, Mindoro Strait, Balabac Strait and any others referred to in the text were included in figure 1.

Reply: These geographical names have been added to Fig. 1 as suggested.

3. Line 32

In some parts of the literature there is a tendency to refer to resonances in terms of their period. However because the angular velocity of resonances (or their frequencies) often form an arithmetic series corresponding 1/4, 3/4, 5/4, etc wavelengths I would recommend replacing periods here by angular velocities (or frequencies), possibly with the periods in brackets for those that need them.

Reply: This comment has been adopted. We have changed the units of period to those of frequency in R1.

4. Line 47-48

It is not that the resonant periods are related but that the two features are parts of the same resonance, the angular velocity of the resonance being determined primarily by the physical properties of the South China Sea.

Reply: We accept this point of view. Accordingly, these statements have been changed to "we investigate the reasons for the GOT to have a strong response around the frequency of one cycle per day and how the physical properties of the SCSB primarily determine the resonances of both the SCSB and the GOT".

5. Line 72: The Numerical Model

It is only in the code availability section that you say that you use the Princeton Ocean Model. I think this needs to be mentioned earlier in the paper as there are many types of ocean model of varying quality. The Princeton Model is well known and is usually acknowledged to be of good quality. However, it includes many options and parameters so, as in any other realistic model study, it is important to show that the version in use can realistically represent the actual tides in the region being studied. For this reason the paper needs an example of the model K1 (and/or O1) tides of the region and comparison with actual tidal observations either in the form of a chart or in the form of comparisons at key tide gauge positions. I realise that Cui et al (2015) did not do this, but if I had refereed their paper I would have made the same point.

Reply: We added the content of the numerical simulation of the tide $K_1$, and the results are compared with the tidal gauges. See the R1, Subsection 2.2.

You could also do with figures showing the flux of tidal energy for both the realistic K1 tidal forcing and with a constant amplitude on the boundary as in your test experiments, to show that the main influx of tidal energy is through the Luzon Strait. If you do not do this it is possible that your analytic model which is based on this assumption is not valid.

Reply: We calculated the tidal energy density of $K_1$ and 0.99 $d^{-1}$, and the results are shown in Figure 2b and 4b. See the R1 to be revised.

6. Line 72: Boundary Condition

More information is needed on the boundary of the model. The domain described in the text seems to be similar to Cui et al (2015) which makes you suspect you used the same code in the same configuration. However, the other paper shows a southern boundary south of the Equator, whereas according to your text the present one is north of the Equator. Why the difference?

Reply: In Cui et al. (2015), the southern boundary is set at 2°S, but in this paper, the southern boundary moves northward to 1.5°N. This change in the southern open boundary does not influence the resonant frequencies but slightly improves the patterns of the amplitude gain and phase change.

Figure 1, which the caption calls the study area, shows only part of the model domain. Why is this? When I first read the introduction and saw this figure, I assumed that this included the model domain with say Luzon Strait as an open boundary and the regions you described as having negligible fluxes as closed boundaries. Your need to make the difference clearer earlier in the paper.

Reply: In R1, we have included an inset in the upper-left corner in Figure 1 showing the entire model domain, and the figure caption is revised as follows: "The South China Sea and its neighbouring area. The contours show the water depth distribution in metres. The blue line B is the mouth cross section of the Gulf of Thailand (GOT). The triangles represent the tidal gauge stations (the full names of

these stations are given in Table 1). The inset in the upper-left corner shows the entire model domains (99-131°E, 1.5-42°N)".

You say that the northern and eastern boundaries as set well away from the South China Sea to limit the effect of the (fixed) boundary condition on the resonances of the South China Sea. However what about the boundaries to the south and east?

Reply: We add the following sentence: "The southern open boundary is set along a latitudinal section of 1.5 °N, which meets the southernmost tip of the Malay Peninsula".

6. Line 64 and 80-90: Real and complex variables

Analysis of waves and oscillating systems tends to be a lot neater and easier to understand when the physical wave is treated as the real part of a function of the type A(x) exp(-i omega t) where A is a complex number and 'i' is the complex i. Then your G and phi are just the amplitude and phase of a complex response function. The appendix would also be a lot shorter if you used complex variables whenever possible.

Reply: Since we use POM in the numerical computations, it is more direct and easier to understand to express the variables as usual functions of x, y and t in the sections on numerical modelling. In the section on theoretical model and in the appendix, we use complex variables as you suggested.

7. Lines 94-96

This is a bit of a mess and needs to be rewritten. From Cui at al. (2015) you know that there are resonant like features which affect both the Gulf of Thailand and the South China Sea. We know that if the Gulf of Thailand was removed, changing the depth of the South China Sea would affect its resonances. Thus what you are really doing here is to see how much changing these resonances affects the resonances of the combined system. (You could have also carried out runs with changed depths in the Gulf of Thailand - in fact I am surprised that you didn't).

What you are not doing here is finding out how the resonances of the South China Sea are affecting (i.e. changing the shape and frequency of) the localised resonances of the Gulf of Thailand.

Reply: According to your comment, we have carried out two additional experiments (numbered Exps. 4 and 5), in which the depths in the GOT are artificially multiplied by 1/2 and 2. The results are added to Table 2 and Figure 3 in R1.

8. Line 100- : The Results

The paper does not specify the geographical location used for figures 2a and 2b. The three SCSB peaks referred to in table 1 refer to the three main peaks of fig 2a. Changing the depth by a factor of 2 seems to change the frequencies by roughly sqrt(2), but this is not discussed.

Reply: (1) In the present study, we do not use specified locations to represent the area of concern; rather, we use the area-mean value of the top 20% amplitude gain to represent the amplitude gain of the corresponding area. This statement was given in the original manuscript and is retained in R1. For clarity, we added this statement to the captions of these figures in R1. (2) This is a good point; thank you. We added the following statement to R1: "It is worth noting that when the depths in the SCSB are artificially changed by factors of 1/2 and 2, the resonant frequencies are roughly changed by factors of $\sqrt{1/2}$ and $\sqrt{2}$, respectively. This indirectly indicates that the quarter-wavelength resonance theory is applicable to the SCS".

Exp 3 shows a second resonance near 1 cycle/day which also seems to have an effect in fig 2b. Changing the depth of the South China Sea will change the frequency of resonances but will not generate new ones. So what is this feature of the South China Sea affecting the Gulf of Thailand?

Reply: In Exp. 3, there is another weaker peak in the SCSB at the frequency of approximately 1.15 $d^{-1}$

(Figure 3a of R1). The peak frequency response may also have an effect on the GOT (plot for Exp. 3 in Figure 3b of R1), which results in a plateau peak of the GOT between 0.5 d⁻¹ and 1.2 d⁻¹. We speculate that this is probably due to the fact that deepening the SCSB may result in a discontinuity of topographic data at the junction with the GOT. However, we are not sure of this speculation, and it is not included in R1.

In the case of fig 2b, representing the Gulf of Thailand response, there are indeed three main peaks matching the peaks in the South China sea plot - but there is also a lot else going on, especially in expt 1 and 2. What resonances are these?

Reply: The second peak of the GOT's response function at the frequency $0.45 \, d^{-1}$ is of some importance. We added the following statement to R1: "In addition, there is a weak response peak at the frequency $0.45 \, d^{-1}$ in the GOT (Exp. 1 in Figure 3b). Since the GOT has a length of 660 km and a mean depth of 36 m, the quarter-wavelength theory gives a resonant frequency of $0.61 \, d^{-1}$. It seems that the peak at $0.45 \, d^{-1}$ is associated with the local regional resonance". Since in this part of study, we use real coastline and topography, which are irregular, the response function also contains some irregular fluctuations, which are difficult to explain.

It should help if the paper illustrated the amplitude and and phase of key resonances. The simplest solution would be to give the amplitude and phase of the solution when forced at the frequencies of the peaks in the response function. A better alternative would be to fit the (complex) response function R(x,w) at a set of w's around each of the resonance peaks to the equation A(x)/(w-w0) + B(x) + C(x)*(w-w0) Where A, B, C, w0 are complex, x is position, w is the (real) angular velocity of the forcing and w0 the estimated (complex) angular velocity of the resonance. A(x) would then be a better approximation to the amplitude and phase of the true resonance. Following the changes, the conclusions to this section need to be rewritten.

Reply: (1) According to this comment, we have added a figure on the distributions of amplitude gains and phase changes to R1 (as Figure 4a). (2) A(x)/(w-w0) + B(x) + C(x)*(w-w0) is a useful equation. Similar equations have been successfully applied to various sea areas to estimate resonant frequency and quality factor Q (e.g., Garrett and Munk, 1971, The age of the tide and the Q of the oceans, DSR; Sutherland et al., 2005, Tidal resonance in Juan de Fuca Strait and the Strait of Georgia, JPO). However, this or similar equations are generally used to fit the observed responses due to sparse sampling in terms of spectrum resolution. For example, in a semidiurnal band, only responses at frequencies of $N_2$, $M_2$, $S_2$ and $K_2$ are generally available. In the present study, the response functions are numerically generated. They are smooth and have fine resolution. Hence, fitting by the above equation would not yield significantly different results. The corresponding author (G. Fang) plans to use this equation in a future study.

9. Line 128

Semantics - the theory is 'applicable' to the Gulf of Thailand but does not explain the enhanced tides around 1 cycle day (although it might explain an enhancement if forced at 2 cycles/day).

Reply: Revised as suggested (the observation shows that semi-diurnal tides are small).

10. Lines 147-160

It would be best if most of this was kept in the appendix. All you really need is eqn 15 and the approximation when r*p_1/p_2 is small.

Reply: Most of the equations have been removed as suggested.

11. Lines 165-185.

I think this needs a little more thought. You should be able to show that the resonances near 0.5 and 2

cycles per day are resonances of the short channel (where cos(beta_2 L_2) is zero) and the one near 1 cycle per day is a resonance of the long channel (where cos(beta_1 L_1) is zero). Then friction reduces the amplitude of the shallow short channel resonances but has little effect on the amplitude of the long channel resonances except within the shallow channel.

Reply: According to your comment, we have added the following paragraph: "If we apply the quarter-wavelength resonance theory to channel 1, we can obtain resonant frequencies of 0.99 $d^{-1}$. If we apply the quarter-wavelength and three-quarter-wavelength resonance theories to channel 2 we can obtain resonant frequencies of 0.61 and 1.84 $d^{-1}$, respectively. Therefore, we can conclude that the major peaks around the frequency of 1.04 $d^{-1}$ in Figure 6 are caused by resonance in channel 1. This indicates that channel 1 plays a determinative role in the two-channel system. Similarly, we can also conclude that the secondary and third peaks around the frequencies of 0.55 and 1.85 $d^{-1}$ in Figure 6 are caused by resonances in channel 2, associated with the quarter-wavelength and three-quarter-wavelength resonances. Although the frequencies of the peaks shown in Figure 6 correspond well with those estimated based on the quarter-wavelength and three-quarter-wavelength theories, there are small discrepancies. This is due to the connection of the two channels. In fact, the resonant frequencies of the two-channel system also depend on the depth ratio of two channels, as shown in Eq. (14). In comparison to channel 2, the secondary, especially the third peak, in channel 1 is much more less significant. This can be explained as follows: The tidal incident wave from the channel 1 partially enters channel 2 across the steep topography at $x = 0$, and here, the rest of the wave is reflected. The reflected wave is superimposed with the incident wave, and tidal resonance occurs around the frequency of 1.04 $d^{-1}$. That is, the steep topography at $x = 0$ acts as a wall for channel 1, which causes the quarter-wavelength resonance to occur in the channel. Furthermore, the steep topography can also block most energy of the wave in channel 2 from entering channel 1. Therefore, the relatively large amplitudes in channel 2 at frequencies around 0.55 $d^{-1}$ and 1.85 $d^{-1}$ are not obvious in channel 1 under the action of friction".

12. Lines 189-195

I suppose my main point here is that the diurnal resonance affecting the Gulf of Thailand in not 'closely related' to that affecting the South China Sea. Instead it is exactly the same resonance.

Reply: Revised as suggested. The statement is changed to "Changing the water depths in the SCSB in our numerical experiments further shows that the resonance of the SCSB has a critical impact on the resonance of the GOT".

13. Appendix

This seems a bit long for the content. I suggest that you cut it down in size, trying to make it more elegant and leaving out some of the obvious steps.

Reply: We simplified the appendix as suggested. Nearly 2/3 of the equations have been removed.

**Modification list**

**(According to Revision #1-Updated manuscript)**

**Page 1.**

In the abstract, we have accepted the suggestion of Dr. Webb.

**Lines 7-10.** The word "period" or "periods" has been replaced with "frequency" or "frequencies", and the corresponding "one day" has also been replaced with "one cycle per day". The words "is approximately" have been replaced with "has a strong response".

**Lines 13-16.** We have added this sentence "We find that the resonant frequency … the main area of the South China Sea".

**Line 20.** Because Figure 1 was replaced with a new figure, the word "red" was changed to "blue".

**Page 2.**

**Lines 2-5 and line 13.** The word "period" or "periods" has been replaced with "frequency" or "frequencies", and the corresponding number has also been changed.

**Lines 20-21.** We have agreed with Dr Webb, so "… the resonant period in the GOT … and SCSB are closely related" has been replaced with '… the GOT to have a strong response… both the SCSB and the GOT.'

**Line 22.** Because the content in section 2 has been changed and we have added section 2.2, the title of section 2 "Numerical methods for estimating the resonant frequency" has been replaced with "The numerical model".

**Line 23.** The title of section 2.1 "Governing equations" has been replaced with "Governing equations and model configuration".

**Lines 24-25.** We have accepted the suggestion of Dr Webb, so we have added the sentence "In this paper, we use the Princeton Ocean Model (POM) for numerical investigation, but we partially modify its code to meet the needs of the present study".

**Line 32.** We added the reference "Webb, 2014".

**Page 3**

**Lines 7-11.** We have added the content of the model configuration. "The computational domain selected is in the range of 99-131 °E and of 1.5-42 °N. … data extracted from navigational charts.'

**Lines 12-28.** We accept the suggestion of Dr Webb, so we newly added section 2.2 about the simulation of the $K_1$ tide. For details, please see section 2.2 of Revision #1.

**Page 4.**

**Line 1.** We have added a new title "3 Numerical methods for estimating the resonant frequency".

**Lines 12-14.** Since the part of the model configuration have already existed in Page 3 lines 7-11, we

deleted it here, and we have added "Through the simulation of the $K_1$ tide … values at the open boundaries are changed as follows", which makes the context more coherent.

**Pages 5-6. Section 4**

In section 4, we have accepted the suggestion of Dr Webb, and this section has been almost rewritten.

**Page 5. Lines 6-10.** We have added this new paragraph "Cui et al. (2015) revealed that both the SCS … be related to the SCS resonance.", which explains why we studied the influence of resonance in South China Sea on the Gulf of Thailand.

**Page 5. Lines11-16.** Since we have carried out two additional experiments 4-5 (numbered Exps. 4 and 5), we added the sentence "In Exp. 4 and 5, we change… SCSB retains the real depths".

**Page 5. Lines16-18.** In order to express the experimental results more clearly, we have accepted the suggestion of Dr. Webb, and we have added this sentence "The results of these six experiments are shown … peak responses are listed in Table 2."

**Page 5. Lines19-25.** This paragraph was newly added. In this paragraph, we compared the spatial patterns of 0.99 $d^{-1}$ and $K_1$ to show that the experimental method is feasible. "Table 2 and Figure 3 show that when real water depths are used … Minor differences are caused by the use of different open boundary conditions."

**Page 6. Lines1-3.** We have accepted the suggestion of Dr Webb, and we have added the analysis "It is worth noting … theory is applicable to the SCSB.".

**Page 6. Lines 5-7.** In these lines, we explained the reasons of producing 1.15 $d^{-1}$ in Exp.3, but we were not sure of the speculation. The details are "The peak frequency response … at the junction with the GOT".

**Page 6. Lines9-13.** Since we have carried out two additional experiments (Exps. 4-5), we added this new paragraph. The details are "Experiments 1-3 suggest that … are still close to one cycle per day".

**Page 6. Lines 14-15.** We have rewritten the lines. "In this experiment, the boundary … as shown in Fig. 3" have been replaced with "In Exp. 6, the mouth boundary … as shown in Fig. 3c".

**Page 6. Lines21-24.** We have added these new lines to explain the reason of producing 0.45 $d^{-1}$ in Exp. 1. The details are "In addition, there is a weak … associated with the local regional resonance."

**Page 6. Lines25-29.** This paragraph is the summery of section 4, and we have moved the summery to the end of section 4.

**Page 6.**

**Lines 31-32.** We have accepted the suggestion of Dr. Webb, the sentence "this theory is not applicable to the GOT" has been replaced with "this theory does not explain the enhanced tides around one cycle per day in the GOT."

**Page 7.**

**Lines 4-5.** We have added the new sentence "This model is quite similar to Webb's (2011) 1-D model, except that we add a forcing at the entrance of the deep channel".

**Lines 16-17.** We have added the new sentence "By eliminating the common … equations as follow" to make the context coherent.

**Page 8.**

**Lines 5-10.** We have accepted the suggestion of Dr. Webb, and we have deleted some equations which have existed in appendix. Besides, we have added the new sentence "When the denominator … in Appendix (Eq. A10)):" to make enunciated coherent

**Page 9.**

**Lines 1-16.** We have accepted the suggestion of Dr. Webb. We have added the new paragraph about the further analysis of the theoretical model results. For details, please see Page 9, lines 1-16.

**Lines 29-30.** "… that the resonance of the GOT is closely related to the resonance in the SCSB" has been replaced with "… that the resonance of the SCSB has a critical impact on the resonance of the GOT"

**Page 9, line 31-page 10, line 2.**

We have added the new sentence "Through the numerical experiments and two-channel model … the main area of the South China Sea"

**Page 10**

**Line 9.** Since we have added the content about simulation of $K_1$ tide which needs the data of the open boundaries, we added the line "The tidal data at the open boundaries are available online at ftp://ftp.oce.orst.edu/dist/tides."

**Page 10, line 10-page 12, line 13.**

This section is the appendix. In this section, we also have accepted the suggestion of Dr. Webb, and we have simplified the appendix. Nearly 2/3 of the equations have been removed.

**Page 13**

**Lines 4-7.** We sincerely thank the topic editor and two referees, and we have added these words "The authors sincerely thank the topic editor … were of great help in improving our study."

**Line 15.** We have added the reference "Egbert G D, Erofeeva S Y…" which is referenced on page 3, line 15.

**Page 14**

**Lines 3-5.** We added two references of Dr Webb, which were of great help and were referenced on Page

3, line 28 and page 7, line 4, respectively.

**Page 15**

**Lines 2-4.** Figure 1 which is on page 20 has been replaced with the new picture, so the caption was changed to "The South China Sea … domains (99-131°E, 1.5-42°N)"

**Lines 5-6.** Figure 2 which is on page 21 was the newly added picture. The caption "Figure 2: Model-produced … flux density vectors (in kW m-1)."

**Lines 8-9.** Figure 3 has been replaced with the new one which is on page 22, and in the caption, we newly have added the sentence "Here, the amplitude gains … given in Table 2" which makes the meaning of Figure 3 clear.

**Lines 10-11.** Figure 4 which is on page 22 was the newly added picture. The caption "Figure 4: (a) Distribution … flux density vectors."

**Pages 16-17.**

Table 1 was the newly added table. For details, please see page 16-17.

**Page 18**

Table 2 has been updated because we have carried out two additional experiments. For details, please see page 18.

**Revision #1 - an updated manuscript**

[revised manuscript text omitted]

---

## Author Response (AR2)

Response to Dr. Wells and Dr D.J. Webb (Referee)

Dear Dr. Wells, we sincerely thanks for your handling our manuscript and for your decision. We have addressed all comments given by Dr. Webb which are helpful in further improving our manuscript. Our response to his comments is given below. In this response, his comments are copied in black, our replies are shown in red, and the following abbreviations are used:

R2 – Revision #2 – the second updated manuscript, which will be submitted together with this response.

This is much improved. I have a few comments below but I will not need to see the paper again.

1. Abstract.

The fact that the gulf is dominated by diurnal tides does not indicate that the resonant frequency of the gulf is close to one cycle per day. It 'might be taken to indicate', but then as the paper shows there can be other reasons.

Reply: We agreed with this comment, and the sentence "… which indicates that the resonant frequency …" has been replaced with "… which might be taken to indicate that the resonant frequency …" in R2.

Later: "of the tidal resonance in the Gulf of Thailand". This should be "... the resonance affecting the Gulf ..."

Reply: This comment has been adopted. "… of the tidal resonance in the Gulf of Thailand …" has been changed to "… of the resonance affecting the Gulf of Thailand …".

2. Introduction

Re: "cycles per day". As a unit Wikipedia recommends 'cpd' (https://en.wikipedia.org/wiki/Cycle_per_second).

Reply: In R2, the units "$d^{-1}$" and "$h^{-1}$" have been replaced with "cpd" and "cph", respectively.

3. Section 2.1

Line 4: Try:

"R indicates the Earth's radius, $\Omega$ its angular velocity, g gravity, H the undisturbed water depth and $\tau$ the linearized bottom friction coefficient".

Reply: This suggestion has been adopted. In R2, "… $R$ indicates the Earth's radius; $\Omega$ refers to the angular speed of the Earth's rotation; $g$ is the gravitational acceleration; $H$ denotes the undisturbed water depth; and $\tau$ represents the linearized bottom friction coefficient" has been replaced with "… $R$ indicates the Earth's radius, $\Omega$ its angular velocity, $g$ gravity, $H$ the undisturbed water depth and $\tau$ the linearized bottom friction coefficient"

4. Line 7: Try:

"The numerical model covers the ocean lying between 99°E, 131°E, 1.5°N and 42°N."

Reply: We accepted this suggestion. In R2, "The computational domain selected is in the range of 99-131 °E and of 1.5-42 °N" has been replaced with "The numerical model covers the ocean lying between 99 °E, 131 °E, 1.5 °N and 42 °N".

5. Section 3.2.

Line 15: Personally I would not expect the response curves of linear model to be affected by the phases

used to force the open boundary - but as you did this you should keep it in.

Reply: Since we only investigate the phase changes (see Eq. (6)), the phase lags specified on the open boundary in principle can be arbitrary. The phase lags generated by random generator as in our work can avoid all or some of the waves to have the same phase at a certain time, which can lead to the simultaneous unreasonably high or low sea levels.

6. Line 20: "In the last cycle of 1024 hours". How long was the model run before this last cycle?

Reply: According to this comment we add a sentence "The model is run for 3×1024 hours (see Cui et al., 2015, Fig.2)." before "In the last cycle of 1024 hours".

7. Section 4. Line 11: Try:

"we conduct six numerical experiments. In experiment 1 ... " or "... In Expt 1 ...". You do not need to repeat yourselves.

Reply: We accepted the suggestion. In R2, we have deleted the words ": Exp. 1 to Exp. 6".

8. Line 19:

You refer to different "resonant frequencies" in the Gulf and the South China Sea. It would be more correct to refer to the peak amplitudes in the two regions - which are close together in frequency and appear to be due to the same underlying resonance.

Reply: We agreed with this comment. In R2, "… the resonant frequency of the SCSB appears at 0.99 $d^{-1}$, while that of the GOT is 1.01 $d^{-1}$ The resonant frequencies of these two areas are basically the same …" has been replaced with "… the frequency corresponding to peak amplitude in the SCSB appears at 0.99 cpd, while that in the GOT is 1.01 cpd. The frequencies corresponding to peak amplitude in the two areas are basically the same …"

9. Page 6, line 28.

I would suggest replacing 'tidal resonance' by 'tidal response' as there is no quarter wave resonance of the Gulf of Thailand.

Reply: Revised as suggested. In R2, "… on the tidal resonance of the GOT …" has been replaced with "… on the tidal response of the GOT …".

10. Figure 3.

The figure has four parts. The caption only refers to two.

Reply: Thank Dr. Webb for pointing out this omission. We have added "… of the SCSB, and response functions of the SCSB (c) and the GOT (d) for different topographies of the GOT and closure of the GOT …" in R2.